# Research on Multi-Hole Localization Tracking Based on a Combination of Machine Vision and Deep Learning

**DOI:** 10.3390/s24030984

**Published:** 2024-02-02

**Authors:** Rong Hou, Jianping Yin, Yanchen Liu, Huijuan Lu

**Affiliations:** 1School of Mechanical and Electrical Engineering, North University of China, Taiyuan 030051, China; b20220106@st.nuc.edu.cn (R.H.); liuyanchenlaoliu@nuc.edu.cn (Y.L.); 2School of Life and Environmental Sciences, GuiLin University of Electronic Technology, Guilin 541004, China; lh_lhj0307@163.com

**Keywords:** machine vision, deep learning, robotic arm, hand–eye calibration, ROS communications, porous disk

## Abstract

In the process of industrial production, manual assembly of workpieces exists with low efficiency and high intensity, and some of the assembly process of the human body has a certain degree of danger. At the same time, traditional machine learning algorithms are difficult to adapt to the complexity of the current industrial field environment; the change in the environment will greatly affect the accuracy of the robot’s work. Therefore, this paper proposes a method based on the combination of machine vision and the YOLOv5 deep learning model to obtain the disk porous localization information, after coordinate mapping by the ROS communication control robotic arm work, in order to improve the anti-interference ability of the environment and work efficiency but also reduce the danger to the human body. The system utilizes a camera to collect real-time images of targets in complex environments and, then, trains and processes them for recognition such that coordinate localization information can be obtained. This information is converted into coordinates under the robot coordinate system through hand–eye calibration, and the robot is then controlled to complete multi-hole localization and tracking by means of communication between the upper and lower computers. The results show that there is a high accuracy in the training and testing of the target object, and the control accuracy of the robotic arm is also relatively high. The method has strong anti-interference to the complex environment of industry and exhibits a certain feasibility and effectiveness. It lays a foundation for achieving the automated installation of docking disk workpieces in industrial production and also provides a more favorable choice for the production and installation of the process of screw positioning needs.

## 1. Introduction

With increasing technological progress, it is inevitable that the automation of industrial production will become more widespread. However, simple industrial robots have been unable to meet the complexity, variability, and unstructured production forms characterizing modern industrial production environments. Machine vision systems use machines instead of human eyes to execute a variety of measurements and judgments. By such means, industrial robots are able to perceive their environments. The addition of industrial cameras to robots is equivalent to adding eyes to the robot, which is then able to recognize workpieces within the range of the camera lens [1,2,3]. Yang et al. [4] from Guizhou University have proposed an intelligent algorithm and system design based on machine vision to achieve accurate recognition and smooth tracking of specific dynamic targets. This system uses a Jetson TX2 as a control core and adopts a convolutional neural network based on the YOLOv3 algorithm to locally realize dynamic target recognition, which is combined with a tracking and obstacle-avoidance algorithm based on fuzzy logic. This approach was used to determine the steering of the target movement and the rotation amplitude, as well as the positions of obstacles, resulting in precise control of the system’s movement and effective obstacle avoidance. Lin et al. [5] have proposed a teleoperated human–computer system based on a six-degrees-of-freedom robotic arm. This system uses a KinectV1 camera, a UR3 robotic arm, and the Microsoft Skeletal Recognition Library to achieve recognition of basic human gestures, as well as real-time tracking of human arm movements by the robotic arm, by means of mapping between the joints of the human arm and those of the robotic arm. The Xinsong Robotics Company (Tianjin, China), in conjunction with the Shenyang Institute of Automation of the Chinese Academy of Sciences, have developed the SR10C-7H welding-specific robot [6], which was equipped with a fixed-machine vision CCD camera in an automated assembly environment to obtain spatial information on workpieces and feed this back to the robotic arm motion actuator through a vision processing system [7]. In Germany, the KUKA company (Augsburg, Germany) has developed a system based on vision and haptic perception for the BMW Mini assembly line; this involves the use of an IBR-iiwa-type industrial robotic arm and an ISRA vision system to achieve rapid recognition of target object shape contours and gripping positions through perception of the surrounding working environment, in order to determine the specific mounting position of workpieces [8]. Xiao et al. [9] have used a magnetic-resonance-compatible robot to increase the operating range of FUS applications and proposed an active tracking and detection method for a robotic arm based on a magnetorheological coordinate system to carry out a robotic-arm-assisted and magnetic-resonance-guided active tracking study of focused ultrasound ablation. Guo et al. [10] have proposed a machine-vision-based screw-hole localization detection method to address the problems of high intensity and low efficiency associated with the manual installation of workpieces. Industrial machine vision systems can replace the human eye to carry out tasks including detection, measurement, identification, and localization [11,12]. In addition, as such systems are able to overcome the inconsistencies associated with human eye standards, they can be customized to meet higher digital standards in industrial quality control and achieve performance levels beyond the limits of the human eye with respect to speed, spectrum recognition, resolution, sensitivity, and reliability. In short, the introduction of machine vision has greatly improved automation and intelligence on production lines [13,14]. Zhao [15] used the SIFT algorithm, Kalman filtering algorithm, and updating the size of the kernel window width to study the algorithm for target tracking of train hooks based on vision, and the results of the video tracking run concluded that the use of target tracking algorithms can be adapted to the working environment of the train-hook picking robot. However, the method is carried out on the basis of being able to recognize the target to be tracked, and the recognition of the target is affected by the actual complex environment, and this part will greatly affect the tracking of the target object, thus affecting the work efficiency. Zhao [16] used a mean shift based algorithm to realize the tracking of the workpiece by establishing a model of the target workpiece and comparing the similarity with the detection region and judging the similarity between the template region and the image of the workpiece to be processed by iterative computation, but the accuracy of this algorithm for the tracking of the workpiece is largely affected by the template features, as well as the light variations.

In industrial production, screw locking is an important process. When locking, it is necessary to accurately identify the position of each threaded hole such that phenomena such as leakage and floating locks may be avoided [17,18]. Traditionally, the positioning of threaded holes has been achieved by recognition with the human eye or by relying on fixtures. Both of these methods involve low efficiency and poor positioning accuracy, and both present difficulties when applied in rapid production [19]. Hadi et al. [20] evaluated the proposed AutoML model using the Case Western Reserve University (CWRU) bearing fault dataset by automated machine learning techniques, using ball bearing faults as an example, and achieved high accuracy, recall, precision, and F1 scores on the test and validation sets, which can help the related industries to reduce the time and cost associated with predictive maintenance. Also, the related work of Hadi et al. on AutoML predictive maintenance fault classification study guided the conduct of the research in this paper. Therefore, in this paper, we consider the assembly process of a production line involving the automatic location and measurement of holes in a wide range of workpieces, as well as the problem that the accuracy of target detection and tracking is largely affected by the environment, and propose a multi-hole localization and tracking method based on machine vision and deep learning, which enables real-time tracking of disk workpieces in a complex on-site environment by classifying and detecting multiple circles on disk workpieces. Localization and tracking of workpieces with multiple holes can then be performed using a robotic arm, thereby improving the efficiency and reliability of the assembly. The primary contributions of this study are as follows:Proposed a method of multi-hole localization and tracking by combining machine vision and deep learning, which reduces the impact of a single method;The YOLOv5 network model is used for data training and detection, which improves the anti-interference ability of target recognition to the environment;The communication between the upper and lower computer adopts ROS communication, which makes the control of the robot have a very good real-time performance;The research ideas in this paper provide important insights for future researchers to improve the research when applied on a large scale.

## 2. Robotic Arm Multi-Hole Localization Tracking System

The robotic arm porous disk localization and tracking system proposed in this paper includes a PC, a robot, a camera, and a porous disk workpiece, as shown in Figure 1.

### 2.1. Working Principle of Porous Disk Positioning

Porous disk positioning is accomplished by means of automatic detection using a machine vision system that includes four links: image acquisition, image processing and data model training, system calibration, and communication between the upper and lower computers. The image acquisition link is responsible for acquiring images with the camera, obtaining high-quality images of the target area through a good distribution of light sources, and storing large numbers of images in the industrial control machine. The PC is responsible for image processing and data model training, processing the acquired images, extracting the eigenvalues of the multiple holes on the disk, obtaining the coordinate information of the multiple holes, and completing the tracking by adjusting the coordinates of the locating holes using the coordinate information of the four peripheral holes and the center hole. System calibration is carried out to convert the extracted coordinate information into the coordinate system of the robotic arm and to control the robotic arm to complete the real-time following of the disk workpiece. Communication between the upper and lower computers is mainly for the purpose of data communication between the PC and the robotic arm industrial control machine.

### 2.2. Localization and Tracking System Realization Process

The camera is fixed at the end of the robotic arm, the image information is acquired and transferred to the PC, and the model is trained on the PC. The acquired porous disk coordinates are first converted from 2D to 3D and are then converted to coordinates under the coordinate system of the robotic arm. The result of the conversion is used to control the robotic arm movement through robot operating system (ROS) communication, such that the localization and tracking of the disk workpiece may be achieved. The flowchart of the tracking system is shown in Figure 2.

## 3. Deep-Learning-Based Target Recognition for Porous Disks

### 3.1. YOLOv5 Deep Learning Framework

Considering the many sources of interference in real-world environments, we use the YOLOv5 (You Only Look Once version 5) deep learning framework to accurately capture target objects in the present study. YOLOv5 is a target detection model for recognizing and localizing objects in videos or images. It is characterized by its fast detection speed, high degree of accuracy, and good ability to detect small objects [21,22,23]. It is an end-to-end deep learning model that detects and localizes targets directly from raw images. It uses convolutional neural networks (CNNs) to learn the features of objects in an image and uses multi-scale prediction and grid segmentation to detect and localize targets. The advantage of YOLOv5 is that it is a single-stage target detection algorithm that works well with different image resolutions. In conclusion, YOLOv5 is an efficient target detection model that can be applied to many different scenarios, including autonomous driving, robot perception, and image analysis. Figure 3 shows a diagram of the core network structure of YOLOv5. Table 1 presents a comparison of model differences between different structures, where each ordinal number corresponds to the corresponding ordinal number in the network diagram of YOLOv5.

The number of convolutional kernels has a significant impact on the performance and effectiveness of the YOLOv5 model. A smaller number of convolutional kernels may lead to insufficient perceptual ability of the model to capture details and features in the image, thus affecting the accuracy of target detection. A larger number of convolutional kernels, on the other hand, may increase the computational complexity and the number of parameters of the model, leading to overfitting of the model or long training time.

In YOLOv5, the number of convolutional kernels is usually set by adjusting the width parameter of the network. A smaller width parameter will reduce the number of convolutional kernels, which is suitable for resource-constrained scenarios but may sacrifice some detection performance. A larger width parameter will increase the number of convolutional kernels and improve the model’s perceptual ability and accuracy, but the computational and storage overhead will be also increased.

Therefore, when using YOLOv5 for target detection, it is necessary to choose the appropriate number of convolutional kernels according to the specific application scenarios and resource constraints. Through experimentation and tuning, the optimal number of convolutional cores can be found to achieve a better detection effect and performance balance.

The four network structures are used to control the depth and width of the network through the parameters depth_multiple and width_multiple in Table 1. In particular, depth_multiple controls the depth of the network, while width_multiple controls the width of the network. Among these four networks, YOLOv5s has the smallest model and the fastest detection speed, but its level of accuracy is relatively low; YOLOv5m and YOLOv5l have successively larger models and higher levels of accuracy; and YOLOv5x has the highest level of accuracy, but also the slowest detection speed. As the targets used in this study were relatively small and, so, high accuracy was required, the YOLOv5x network structure was selected for training and detection.

### 3.2. Model Training

Considering the subsequent development and research, the camera selected in this paper is the Realsense D415 camera, which has a high resolution and can measure the depth (the minimum depth can be measured to 35 cm). Since the overall size of the robot arm in this paper is less than 35 cm, no research has been added to this part. Therefore, images with a resolution of 640 × 480 were collected for training. After a certain number of images (about 2000) were collected by the camera, a dataset suitable for YOLOv5 training was generated by labeling the targets of the images individually, for the purposes of training and testing. The precision of object detection is mainly measured by precision, recall, F1 Score, PR_curve, mAP, and other indexes. Figure 4, Figure 5 and Figure 6 show the indicators after model training, and the recognition accuracy of the target object is measured by analyzing relevant indicators.
(1)$$Precision=\frac{TP}{TP+FP}$$
(2)$$Recall=\frac{TP}{TP+FN}$$
(3)$$F1Score=\frac{2\times Precision\times Recall}{Precision+Recall}$$
where *TP*: true positive; *FP*: false positive; *FN*: false negative. And the value of F1 ranges between 0 and 1, with 1 being the best and 0 being the worst.

For the PR_curve in Figure 5, the PR_curve is the relationship curve between precision and recall, where P is mAP (mean average precision). It is expected that both P and R can obtain 1 under the premise of high accuracy. We sought to detect all categories with the highest possible degree of accuracy; therefore, when the curve is close to the point (1, 1)—that is, when the area of the mAP curve is close to 1—the results are better. As can be seen from the figure, the area of the mAP curve trained for the purposes of the present study was almost 1. 

The loss function is used to measure the degree of difference between the predicted value and the real value of the model, which largely determines the performance of the model.

For each training evaluation index in Figure 6, the horizontal coordinate represents the number of training rounds, and for the six indexes in Figure 6a, it reflects the relationship between the presumed and verified mean value of the loss function. The smaller the mean value of the loss function in the vertical coordinate, the more accurate the target detection is, and the loss in the figure decreases to 0, so the training effect is better. For the four indicators in Figure 6b, they are the indexes to measure the quality of the detection accuracy of the target. The fluctuation of the accuracy and recall rate reflects the training effect. The smaller the fluctuation, the better the effect. It can be seen that the fluctuation in the figure is not large, so the training effect is better.

The YOLOv5 object detection pseudocode is as Algorithm 1:**Algorithm 1.** The YOLOv5 object detection pseudocodeInput: imagesOutput: target detection information  1.  Load pre-trained YOLOv5x model: yolov5x.load_pretrained_model()  2.  Define image preprocessing steps: transform = transforms.Compose ([                        transforms.ToTensor()])  3.  Input images: image = transform(Image.open(image_path)).unsqueeze(0)  4.  Input the image into the YOLOv5x model:     with torch.no_grad():        model.eval()        predictions = model(image)  5.  Analysis of prediction results:  boxes = predictions[..., :4] # Extract the bounding box coordinate information  scores = predictions[..., 4] # Extract the confidence  class_probs = predictions[..., 5:] # Extraction class probability  6.  Non-maximum suppression (NMS):  Selected_boxes = torch.ops.torchvision.Nms (boxes, scores,  iou_threshold = 0.5)  7.  Return the final result:  final_boxes = boxes[selected_boxes]  final_scores = scores[selected_boxes]  final_class_probs = class_probs[selected_boxes]

### 3.3. Target Recognition Detection

After training the dataset marked with labeling by the YOLOv5x network structure, we used the best weights obtained during training to carry out detection of porous disk workcraft and obtain the porous disk pixel coordinates. The identification and detection results are shown in Figure 7, from which it can be seen that the confidence of recognition reached a level above 90%.

## 4. Coordinate Transformation and Calibration

Camera imaging is the mapping of real-world objects to an imaging plane. This principle is used for small-hole imaging models [24], an example of which is shown in Figure 8. As the coordinate information obtained by image processing is in the form of pixel coordinates, in order to perform target tracking operations, the pixel coordinates are converted into world coordinates.

### 4.1. Coordinate Conversion

The world coordinate system is based on the three-dimensional spatial coordinates of the real world. It is therefore a three-dimensional coordinate system, as indicated by $O_{W}-X_{W}Y_{W}Z_{W}$ in Figure 8.

The camera coordinate system is based on the principle of lens imaging and involves the presentation of world coordinates to the camera. It is also a three-dimensional coordinate system, as indicated by $O_{C}-X_{C}Y_{C}Z_{C}$ in Figure 9.

The image coordinate system is projected onto the screen, and the new coordinate system established does not include high-profile information. It is a two-dimensional coordinate system, as indicated by o−xy in Figure 9.

Pixel coordinates are pictures formed by discrete sampling of projection image coordinates. They are a two-dimensional coordinate system, as indicated by uv in Figure 9.

The conversion of world coordinates to pixel coordinates involves three transformations [25,26,27], as follows: conversion from world coordinates to camera coordinates, conversion from camera coordinates to image coordinates, and conversion of pixel coordinates to world coordinates. The conversion relationships for each coordinate system may now be stated.

The conversion between world coordinates and camera coordinates is as follows:

The coordinates of point P in the camera coordinate system may now be obtained as follows:(4)$$\left[\begin{array}{c} X_{C}\\ Y_{C}\\ Z_{C} \end{array}\right]=R\left[\begin{array}{c} X_{w}\\ Y_{w}\\ Z_{w} \end{array}\right]+T\Rightarrow \left[\begin{array}{c} X_{C}\\ Y_{C}\\ Z_{C}\\ 1 \end{array}\right]=\left[\begin{array}{cc} R & T\\ \vec{0} & 1 \end{array}\right]\left[\begin{array}{c} X_{w}\\ Y_{w}\\ Z_{w}\\ 1 \end{array}\right]$$
where *R* is a 3 × 3 matrix and *T* is a 3 × 1 matrix.

The conversion between the camera coordinate system and the image coordinate system is as follows:

The relationship between the two groups of similar triangles, $\Delta ABO_{C}\sim \Delta oCO_{C}$ and $\Delta PBO_{C}\sim \Delta pCO_{C}$, can be obtained as follows:(5)$$Z_{C}\left[\begin{array}{c} x\\ y\\ 1 \end{array}\right]=\left[\begin{array}{cccc} \begin{array}{c} f\\ 0\\ 0 \end{array} & \begin{array}{c} 0\\ f\\ 0 \end{array} & \begin{array}{c} 0\\ 0\\ 1 \end{array} & \begin{array}{c} 0\\ 0\\ 0 \end{array} \end{array}\right]\left[\begin{array}{c} X_{C}\\ Y_{C}\\ Z_{C}\\ 1 \end{array}\right].$$

The conversion between pixel coordinates and world coordinates is carried out as follows:(6)$$Z_{C}\left[\begin{array}{c} u\\ v\\ 1 \end{array}\right]=\left[\begin{array}{ccc} \frac{1}{dx} & 0 & u_{0}\\ 0 & \frac{1}{dy} & v_{0}\\ 0 & 0 & 1 \end{array}\right]\left[\begin{array}{ccc} \begin{array}{cc} \begin{array}{c} f\\ 0\\ 0 \end{array} & \begin{array}{c} 0\\ f\\ 0 \end{array} \end{array} & \begin{array}{c} 0\\ 0\\ 1 \end{array} & \begin{array}{c} 0\\ 0\\ 0 \end{array} \end{array}\right]\left[\begin{array}{cc} R & T\\ \vec{0} & 1 \end{array}\right]\left[\begin{array}{c} X_{w}\\ Y_{w}\\ Z_{w}\\ 1 \end{array}\right]=\left[\begin{array}{cccc} \begin{array}{c} f_{x}\\ 0\\ 0 \end{array} & \begin{array}{c} 0\\ f_{y}\\ 0 \end{array} & \begin{array}{c} u_{0}\\ v_{0}\\ 1 \end{array} & \begin{array}{c} 0\\ 0\\ 0 \end{array} \end{array}\right]\left[\begin{array}{cc} R & T\\ \vec{0} & 1 \end{array}\right]\left[\begin{array}{c} X_{w}\\ Y_{w}\\ Z_{w}\\ 1 \end{array}\right],$$
where $\left[\begin{array}{cccc} \begin{array}{c} f_{x}\\ 0\\ 0 \end{array} & \begin{array}{c} 0\\ f_{y}\\ 0 \end{array} & \begin{array}{c} u_{0}\\ v_{0}\\ 1 \end{array} & \begin{array}{c} 0\\ 0\\ 0 \end{array} \end{array}\right]$ is the internal reference of the camera and $\left[\begin{array}{cc} R & T\\ \vec{0} & 1 \end{array}\right]$ is the external parameters of the camera.

Equations (5) and (6) may now be deduced from $Z_{C}\left[\begin{array}{c} u\\ v\\ 1 \end{array}\right]=M_{c}\left[\begin{array}{cc} R & T\\ \bar{0} & 1 \end{array}\right]\left[\begin{array}{c} X_{w}\\ Y_{w}\\ Z_{w}\\ 1 \end{array}\right]$:(7)$$Z_{C}{M_{C}}^{-1}\left[\begin{array}{c} u\\ v\\ 1 \end{array}\right]=RT,$$
(8)$$\left[\begin{array}{c} X_{w}\\ Y_{w}\\ Z_{w} \end{array}\right]=R^{-1}({M_{C}}^{-1}Z_{C}\left[\begin{array}{c} u\\ v\\ 1 \end{array}\right]-T),$$
where *M_C_* is the camera’s intra-insinuity matrix.

After obtaining the internal and external parameters of the camera, the world coordinates of the object are derived by working backward through the relationship between pixel coordinates and world coordinates.

### 4.2. Calibration Method

Camera calibration means that objects captured by the camera are in a three-dimensional world coordinate system. Camera imaging converts the three-dimensional camera coordinate system to the two-dimensional image coordinate system. Calibration allows for an approximate estimation of the transformation matrix and distortion coefficient. In unstructured and dynamic scenarios, hand–eye calibration is required for perception and localization [7,8,9,10]. The purpose of hand–eye calibration is to connect the perception module with the planning module of the robotic arm, such that the robotic arm can dynamically localize to the target position for grasping through the “eye”. The eye of the hand–eye calibration is on the hand, so that “eye in hand” refers to the camera fixed to the end of the robotic arm, as shown in Figure 10.

The hand–eye calibration coordinate system is represented by four coordinate systems: The base coordinate system of the robotic arm, the end coordinate system of the robotic arm (end), the camera coordinate system (camera), and the calibration plate coordinate system (board).

The eye mainly calibrates the transformation matrix of the end of the camera and the robotic arm—that is, the transformation matrix Mendcamera from the coordinate system at the end of the robotic arm to the camera coordinate system. The implementation method is as follows:The calibration plate is secured in a fixed position, such that it does not move;The end of the robotic arm is moved to take multiple pictures of the calibration plate from different angles.

It is known that each image may be expressed as follows:(9)Mendcamera=Mboardcamera×Mbaseboard×Mendbase,
where Mboardcamera, the calibration plate-to-camera transformation matrix (i.e., target pixel coordinates to camera coordinates), is obtained by solving for the captured calibration plate picture; Mendbase is obtained from the end position parameter of the robot arm; and Mendboard is an unknown quantity. This transformation matrix is the same for each set of pictures as the calibration plate is fixed in one position and does not move at any time in the process.

The deformation can now be expressed as:(10)Mbaseboard=Mboardcamera−1×Mendcamera×Mendbase−1.

From this, we obtain:(11)Mboardcamera2×Mboardcamera1−1×Mendcamera=Mendcamera×Mendbase2−1×Mendbase1.

Solving the system of equations AX=XB gives a value of *X* as the transformation matrix Mendcamera from the end of the robotic arm to the camera coordinate system, where
(12)A=Mboardcamera2×Mboardcamera1−1, B=Mendbase2−1×Mendbase1.

Grasping of the target after hand–eye calibration may now be expressed as:(13)Pb=Mendbase×Mendcamera×Pc,
where *P_c_* represents the vector of the clamped target in the camera coordinate system, which is calculated from the depth information and the pinhole camera model, and *P_b_* represents the vector of the target in the basic coordinate system. The inverse kinematics of the manipulator can now be called on through *P_b_* to actually complete the grasping and positioning tasks.

### 4.3. Coordinate Mapping Reasoning

Coordinate mapping of each node was deduced for the six-degree-of-freedom robotic arm, and the analysis diagram is shown as Figure 11.

Camera to end:(14)$$L_{0}={\left(dx_{0},dy_{0},dz_{0}\right)}^{T}$$
(15)$$X_{1}=X_{0}+L_{0}$$
where $dy_{0}$ = 0.

Camera to first joint:(16)$$L_{1}={\left(dx_{1},dy_{1},dz_{1}\right)}^{T}$$
where $dy_{1}$ = 0.

First joint coordinate system:(17)$$X_{2}=X_{1}+L_{1}$$
(18)$$X_{2}=X_{0}+L_{1}={\left(dx_{0}+dx_{1},0,dz_{0}+dz_{1}\right)}^{T}$$

First joint to second joint (rotation around the y-axis, y is constant, and the y-axis is omitted from the figure):(19)$$M_{0}=\left[\begin{array}{ccc} {cos\theta }_{1} & 0 & {sin\theta }_{1}\\ 0 & 1 & 0\\ {-sin\theta }_{1} & 0 & {cos\theta }_{1} \end{array}\right]$$
(20)$$L_{2}={\left(l_{1},0,\;0\right)}^{T}$$
where $l_{1}$ is the rod length between the first and second joints.

Second joint coordinate system:(21)$$X_{3}={M_{0}X}_{2}+L_{2}$$
(22)$$X_{3}=\left[\begin{array}{ccc} {cos\theta }_{1} & 0 & {sin\theta }_{1}\\ 0 & 1 & 0\\ {-sin\theta }_{1} & 0 & {cos\theta }_{1} \end{array}\right]X_{2}+L_{2}$$
(23)$$X_{3}=\left[\begin{array}{ccc} {cos\theta }_{1} & 0 & {sin\theta }_{1}\\ 0 & 1 & 0\\ {-sin\theta }_{1} & 0 & {cos\theta }_{1} \end{array}\right]\left[\begin{array}{c} x_{0}+dx_{0}+dx_{1}\\ y_{0}\\ z_{0}+dz_{0}+dz_{1} \end{array}\right]+\left[\begin{array}{c} l_{1}\\ 0\\ 0 \end{array}\right]$$
(24)$$M_{1}=\left[\begin{array}{ccc} {cos\theta }_{2} & 0 & {sin\theta }_{21}\\ 0 & 1 & 0\\ {-sin\theta }_{2} & 0 & {cos\theta }_{2} \end{array}\right]$$
(25)$$L_{3}={\left(l_{2},0,\;0\right)}^{T}$$

Third joint coordinate system:(26)$$X_{4}={M_{1}X}_{3}+L_{3}$$

Fourth joint coordinate system:(27)$$X_{5}={M_{2}X}_{4}+L_{4}$$

Variables: $\theta_{1},\theta_{2},\theta_{3},X_{0}$, where $\theta_{1}$ is angle difference between the first and second joints.

Base coordinate system:(1)Projection:

Initial angle of the base joint: $\theta_{4}$

Angle of point P in the $X_{5}$ coordinate system with respect to the x-axis: $\alpha_{4}$

Angle of connecting rod to the horizontal in joints III and IV: $\alpha_{level\;surface}$



(28)
$$L_{5}={\left({0,0,z_{5}+l}_{5}\right)}^{T}$$


(29)
$$X_{projection}=\left[\begin{array}{ccc} {cos\alpha }_{level\;surface} & 0 & {sin\alpha }_{level\;surface}\\ 0 & 1 & 0\\ 0 & 0 & 0 \end{array}\right]\times X_{5}$$



Variables: $\theta_{1},\theta_{2},\theta_{3},X_{0},\alpha_{levelsurface}$.


(2)Rotate $\theta_{4}$ around the z-axis:

(30)
$$X_{base}=\left[\begin{array}{ccc} {cos\theta }_{4} & {sin\theta }_{4} & 0\\ {-sin\theta }_{4} & {cos\theta }_{4} & 0\\ 0 & 0 & 1 \end{array}\right]\times X_{projection}+L_{5}$$



Variables: $\theta_{1},\theta_{2},\theta_{3},\theta_{4}{,X}_{0},\alpha_{level\;surface}$.

The relation obtained by the final inference is the relation of the calibration method described in the previous section.

## 5. Communication and Control of the Host Computer and Robotic Arm

The coordinate information obtained from the image processing part needs to be transferred to the robotic arm for control. In the present study, a topic communication mechanism was used to complete communication between the host computer and the robotic arm. Topic communication is one of the most commonly used communication methods based on the publish–subscribe model in ROS communication, which is mostly used in real-time cyclic data transmission scenarios involving less logic processing. The topic communication model is shown in Figure 12.

In the present study, this form was used to complete data transmission for real-time adjustment and positioning. In the following section, the direct communication between the ROS and the real robotic arm, as well as the control achieved, is specifically introduced.

### 5.1. Robotic Arm Hardware Configuration

The joints of the Dofbot robotic arm use the DS-SY15A bus servo manufactured by De Sheng Model Technology Co., Ltd., located in Guangdong, China, as a power source, which is controlled by serial commands. The development board leads an IO port to connect to the servo, and the servo can be connected in series through the dual interfaces. This reduces the occupation of the development board’s interface and enables reading of the angle, temperature, and voltage.

The servos have built-in driver circuits that send the correct data frames to the servos through the serial bus, such that control of servo rotation may be achieved. Six servos are connected to each other in series through the serial bus to achieve communication with the development board. The overall schematic of the Dofbot robotic arm is shown in Figure 13.

The Dofbot arm is driven by a multi-functional driver board. The driver board houses a Jetson Nano master control that runs Linux, which calls on the ROS MoveIt package for path planning and sends control commands to the driver board to make the six-degrees-of-freedom robotic arm move according to the desired path.

### 5.2. Control Algorithms

#### 5.2.1. Robot Kinematics

The robotic arm used in this paper is the Dofbot robotic arm, whose overall dimensions are shown in Figure 14 below:

Dofbot’s coordinate system is established as Figure 15: 

The red axis is the x-axis, the green axis is the y-axis, and the blue axis is the z-axis, where the bottom coordinate system is the base coordinate system.

$\alpha_{i-1}$: The angle between ${\hat{Z}}_{i-1}$ and ${\hat{X}}_{i}$ as viewed in the direction ${\hat{X}}_{i-1}$;

$a_{i-1}$: The distance between ${\hat{Z}}_{i-1}$ and ${\hat{X}}_{i}$ as viewed in the direction ${\hat{X}}_{i-1}$($a_{i}>0$);

$\theta_{i}$: The angle between ${\hat{X}}_{i-1}$ and ${\hat{X}}_{i}$ as viewed in the direction ${\hat{Z}}_{i}$;

$d_{i}$: The distance between ${\hat{X}}_{i-1}$ and ${\hat{X}}_{i}$ as viewed in the direction ${\hat{Z}}_{i}$;

In this paper, the D-H parameter method is used to describe the geometric relationship between the connecting rod and the joint in the serial link of Dofbot manipulator. The D-H parameters of Dofbot are show as Table 2.

By solving the inverse kinematics of the robotic arm and analyzing its inverse kinematics, control of the robotic arm is achieved. Inverse kinematics refers to solving the angles of each rotational joint of a given robotic arm’s end pose, which is a homogeneous transformation matrix.

The dofbot.dofbot_getIK(urdf_file, targetXYZ, targetRPY, outjoints) function is used for inverse kinematics calculation of Dofbot, which calculates the angles of each joint reaching the target point, where

@param urdf_file: model file path;

@param targetXYZ: target position;

@param targetRPY: target attitude;

@param outjoints: target point joint angle.

#### 5.2.2. MoveIt Guides Path Planning

The MoveIt function package was used for path planning in the ROS robotics framework. The key node in MoveIt is move_group; its system structure is shown in Figure 16.

The user provides endpoint target information to move_group through move_group_interface. Then, move_group carries out the path planning and releases the spatial trajectory from the waypoint information of each joint outward through JointTrajectoryAction; the robotic arm motion control node receives the information from JointTrajectoryAction, then controls the robotic arm’s various joints to carry out the motion accordingly.

### 5.3. Realization Process

#### 5.3.1. Secondary Development of Dofbot

Based on the urdf model of the Dofbot robot arm, a MoveIt function package was generated using the MoveIt configuration assistant for path planning.

#### 5.3.2. Robotic Arm Control Node

A robotic arm control node, armCtrl, was created to receive waypoints published by the move_group node, in order to control the movement of the robotic arm and to read back the current state of the robotic arm. When a valid JointTrajectoryAction is received, the node sends control commands to the robotic arm according to the path planned by move_group at a frequency of 10 Hz, controlling the robotic arm to perform a specific movement through the interface in the Arm_Lib class. At the same time, the armCtrl node reads the position information for each joint of the robotic arm at a frequency of 12.5 Hz, and publishes it with the topic “Joint States Topic”. The move_group node can subscribe to this topic to monitor the current position of the robotic arm in real time and realize closed-loop control.

## 6. Experimental Results

### Tracking Porous Disk Experiments

After a small laser module is installed above the camera at a horizontal distance of 0.25 cm and a vertical distance of 1.7 cm from the midpoint, a porous disk with a diameter of 14 cm is moved on the magnetic plate with grid lines (each grid is 5 cm) in the neighborhood of point 42 (see Figure 17a) for the numbering of each neighborhood to carry out target tracking and detection. The following result is reflected by the laser drop point. The test results are shown as Figure 17.

It can be seen from the results that the accuracy of target detection and tracking is relatively high, and the error between the moving laser drop point and the 42-point drop point is relatively small. It can be seen from Figure 18e,f that the laser drop point is tilted upward because the height at point 17 and point 19 is already at the limit of the mechanical arm’s extension, so the end is in the upward attitude, and so, the laser drop point is tilted upward. As can be seen in Figure 18c,d, the laser drop point is lower because the camera and the fixed end of the mechanical arm have a certain weight, and the mechanical arm needs to overcome gravity to work, so the drop point is lower.

The paper also conducts tracking tests for multiple directions and multiple moving distances to further measure tracking errors. After model training, coordinate conversion, and communication between the upper and lower position machines, the location tracking experiment of the porous disk is carried out. The tracking accuracy of the porous disk (diameter 14 cm) was examined according to the horizontal displacement, vertical displacement, and a combined horizontal and vertical displacement, and data points were represented on the data laser point location map by small colored dots (0.15 cm in diameter), as follows:

Red: initial point;

Dark blue: the laser landing point after moving 5 cm;

Green: laser landing point after moving 10 cm;

Light blue: the laser landing point after moving 15 cm;

Yellow: the laser drop point after moving 20 cm.

Tracking of horizontal displacements was conducted (5 cm to the left each time, with detection of error).

(1) The following figure shows five sets of data regarding leftward displacement and the position diagram of the laser drop point on the disk (showing an enlarged drop point position map), where the large circle in Figure 19a,b is the circle within the red rectangular box of Figure 19f (with a diameter of 12.3 mm). All the following laser drop-point data position maps are graphed similarly.

(2) The following figure shows five sets of data regarding the rightward displacement and the position of the laser landing point. The figures are as described above (Figure 20).

It can be determined, from the left and right movement data, that the laser landing points for horizontal tracking were relatively concentrated and that the error of horizontal tracking was less than 1.5 mm.

Next, the tracking of vertical displacements was also conducted (5 cm per vertical movement each time, with detection of error).

(1) The following figure shows five sets of data regarding upward displacement and the position of the laser drop point. The figures are as described above (Figure 21).

Through the upward movement of these groups of laser drop-point position maps, it can be seen that these drop points were relatively scattered; however, they were also located around the initial point, and the error was less than 5 mm. This is because, when the mechanical arm moves upward, the suspended mechanical arm has to overcome gravity to work due to its weight, and so, there is some shaking. For this reason, the error is larger than with the horizontal displacement.

(2) The following figure shows five sets of data regarding downward displacement and the position of the laser landing point. The figures are as described above (Figure 22).

From the above sets of laser landing-point position diagrams, it can be seen that the landing points were relatively concentrated, and the error is less than 3 mm.

Next, combined horizontal and vertical motion tracking was conducted (5 cm horizontally and 5 cm vertically each time, with detection of error).

(1) The following figure shows five sets of data and laser drop-point location diagrams with both left and upward displacement. The figures are as described above (Figure 23).

(2) The following figure shows five sets of data and laser drop-point location diagrams that are both displaced to the right and upward. The figures are as described above (Figure 24).

(3) The following figure shows five sets of data and laser drop-point location maps that are both left and downward. The figures are as described above (Figure 25).

(4) The following figure shows five sets of data and laser drop-point location diagrams that are both displaced. The figures are as described above (Figure 26).

It can be seen from the above tracking laser drop location graphs combining the horizontal and vertical displacements that the laser drops were relatively concentrated. When this finding was combined with the analysis of the drop data in Appendix A, it could be concluded that, when the robotic arm was tracking the circular holes, the error was within 1.5 mm.

In summary, the multi-hole tracking and localization method proposed in this paper achieved accuracy within a 1.5 mm level of error, despite the limitations of the hardware used. It may be regarded, therefore, as a good foundation for automated assembly in various industries.

## 7. Conclusions

Aiming at the problem of large-scale automatic measurement and automatic assembly of the workpiece in a complex field environment, the system improves the anti-interference ability of the workpiece environment through the combination of machine vision and YOLOv5 deep learning. The robot arm is controlled by ROS topic communication in real time, and the porous disk is detected. The coordinates are transformed for real-time positioning and following operations, and the robot arm is controlled to track the hole position of the disc. This method has relatively high detection accuracy for small objects and small following error, which breaks the fixed and repetitive work of industrial robots, meets the needs of workpiece positioning on industrial production lines, provides a good prospect for the automation and intelligence of assembly lines, and also provides important ideas and insights for the assembly process of positioning small objects in industry.

However, due to the influence of hardware conditions and algorithms, the research described in this paper still has the following limitations, which need to be addressed in future studies:The control accuracy of the servo was 1°. This was inadequate, as it required that some values be discarded when positioning. Some values should be discarded when positioning, and the steering gear with higher accuracy can be replaced by further research.The training performance of the model was relatively large, and the training time was relatively long. Subsequently, it can be used for detection after training by optimizing algorithms or selecting cloud servers.

## Figures and Tables

**Figure 1 sensors-24-00984-f001:**
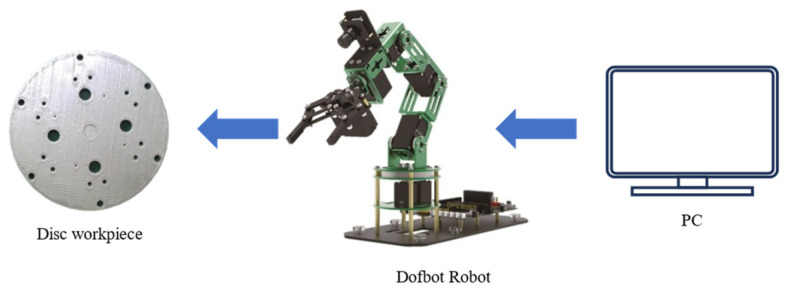
Schematic diagram of robotic arm multi-hole localization tracking system.

**Figure 2 sensors-24-00984-f002:**
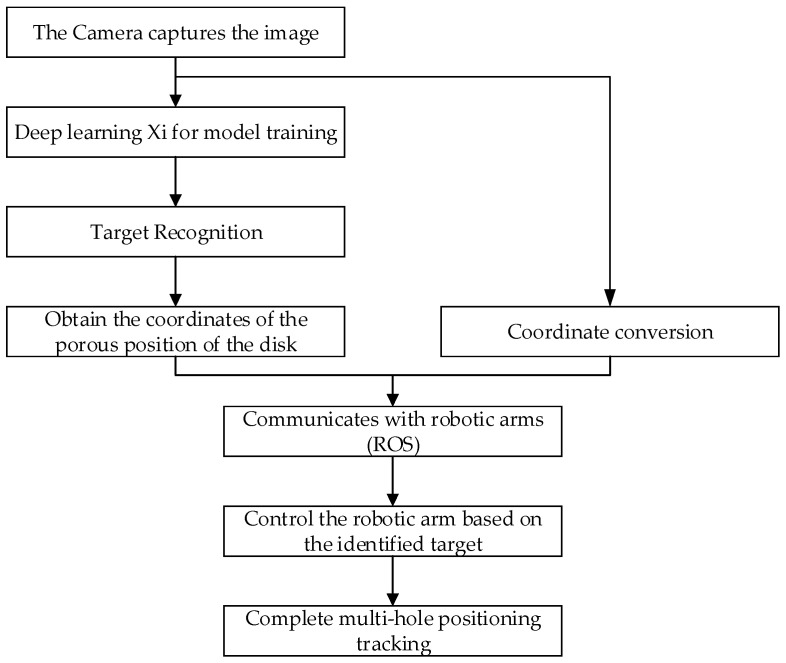
Flow chart of tracking system.

**Figure 3 sensors-24-00984-f003:**
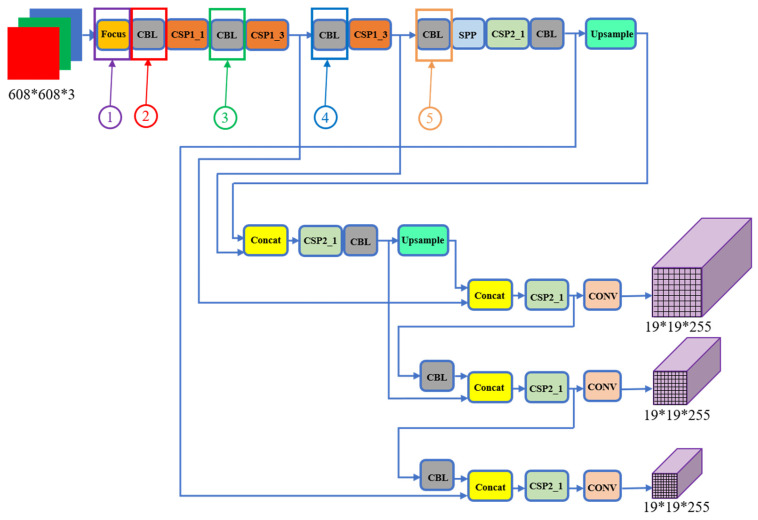
YOLOv5 core section.

**Figure 4 sensors-24-00984-f004:**
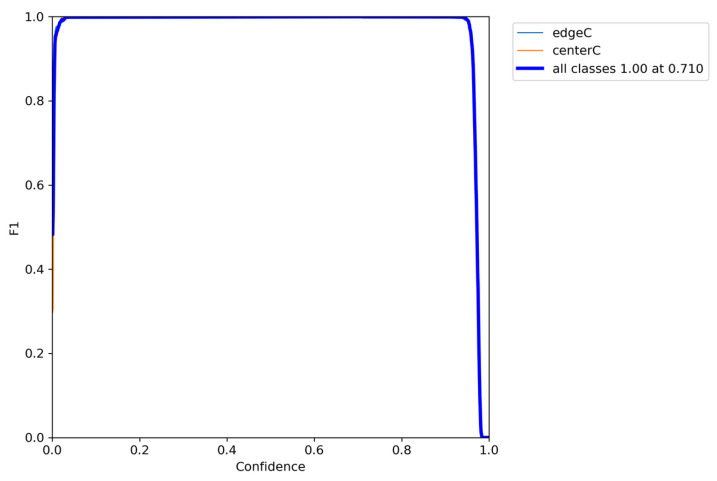
F1_confidence curve.

**Figure 5 sensors-24-00984-f005:**
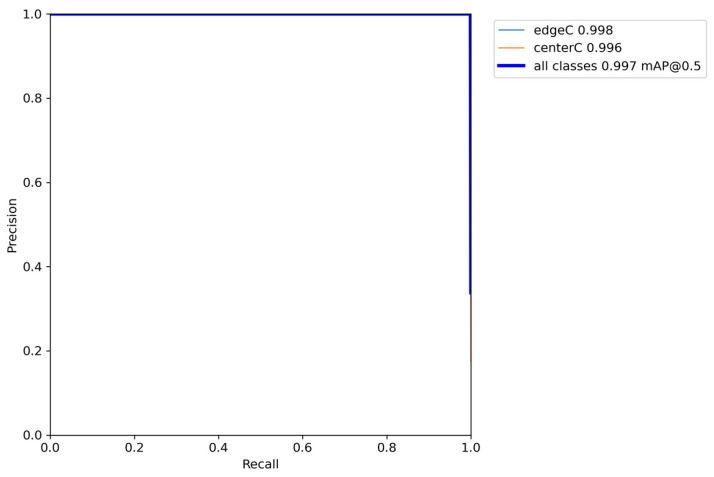
PR_curve.

**Figure 6 sensors-24-00984-f006:**
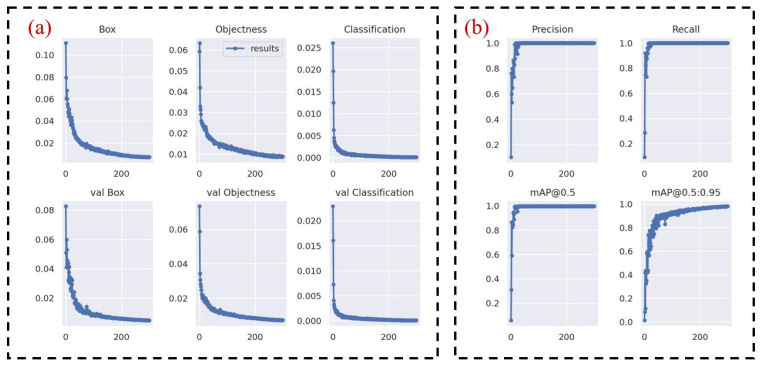
Training each evaluation index: (**a**) training evaluation index; (**b**) detection accuracy of the target.

**Figure 7 sensors-24-00984-f007:**
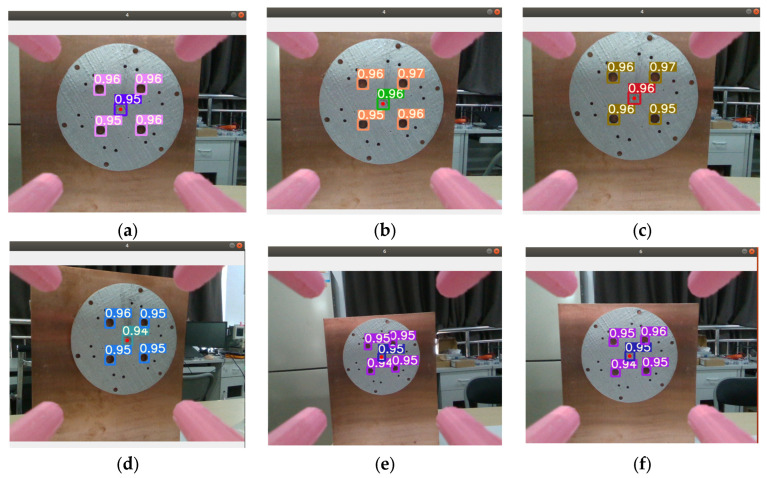
Training each evaluation index: (**a**–**l**) are the results of tests conducted at a distance; (**j**–**l**) are the results of the test at the closer location.

**Figure 8 sensors-24-00984-f008:**
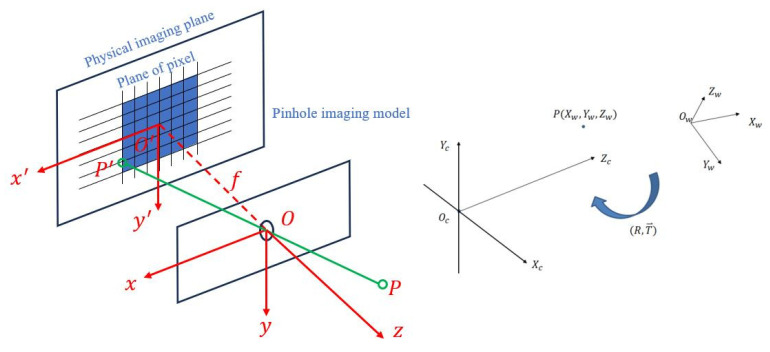
Camera imaging model.

**Figure 9 sensors-24-00984-f009:**
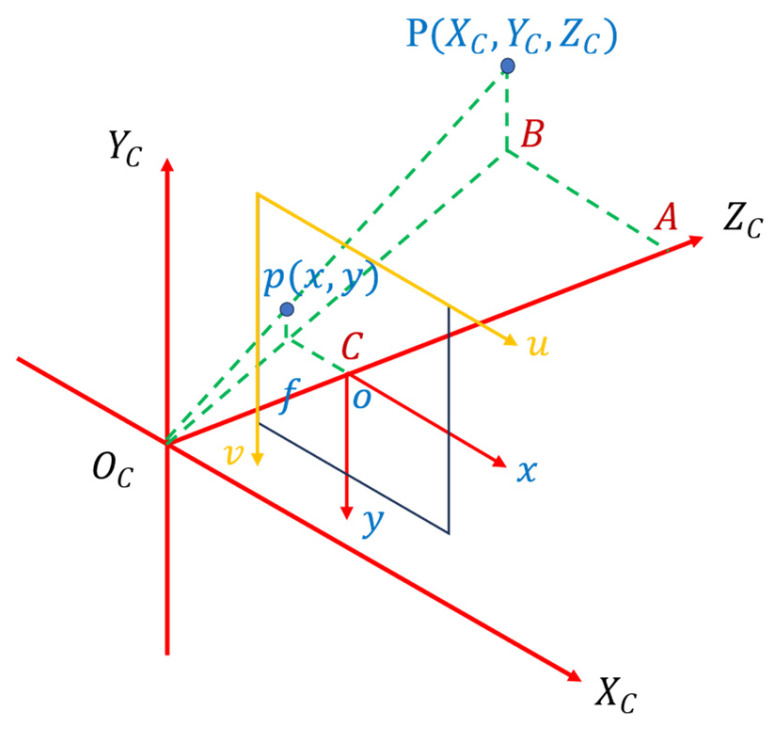
Schematic diagram of camera and image conversion.

**Figure 10 sensors-24-00984-f010:**
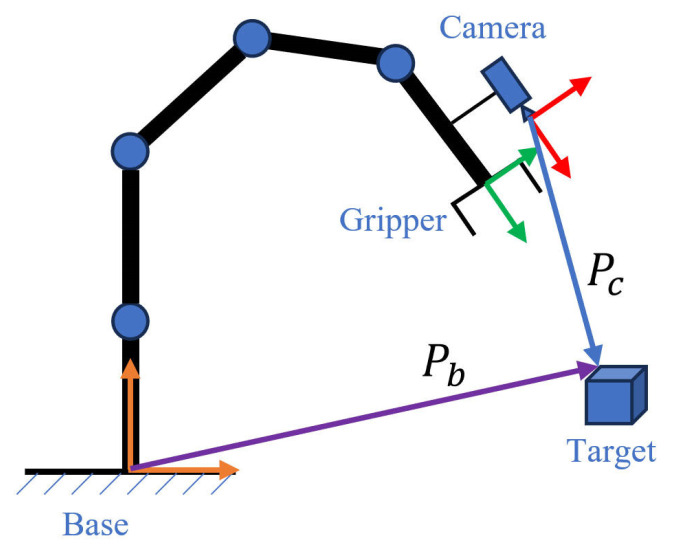
Schematic of the “eye in hand” of a robotic arm.

**Figure 11 sensors-24-00984-f011:**
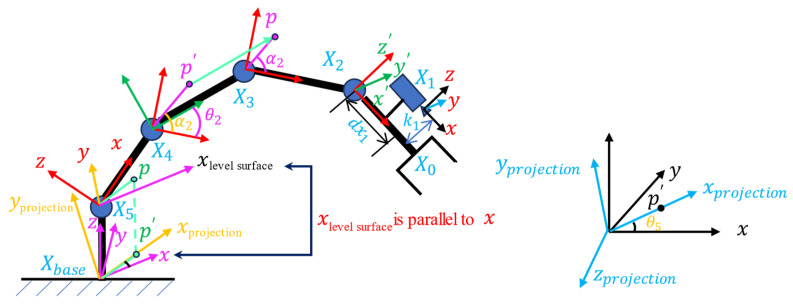
Coordinate mapping of each node.

**Figure 12 sensors-24-00984-f012:**
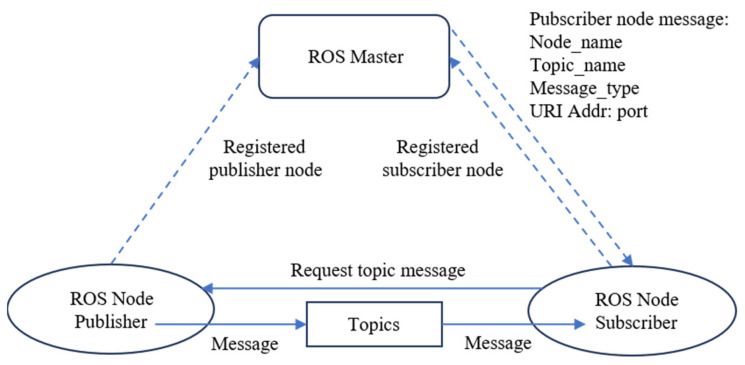
ROS topic communication model.

**Figure 13 sensors-24-00984-f013:**
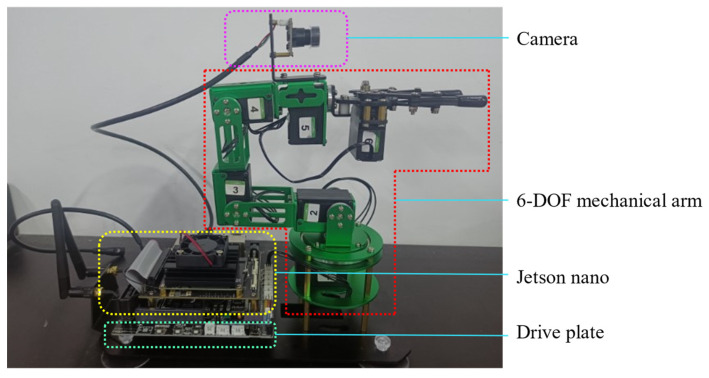
Schematic diagram of the robotic arm.

**Figure 14 sensors-24-00984-f014:**
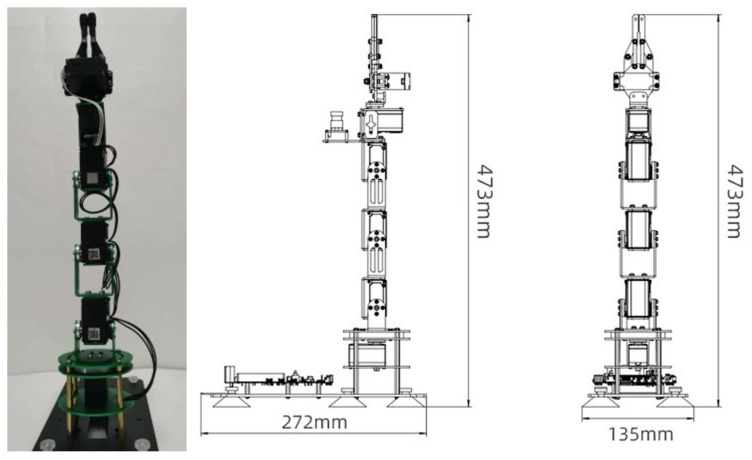
Dimensional drawing of robotic arm.

**Figure 15 sensors-24-00984-f015:**
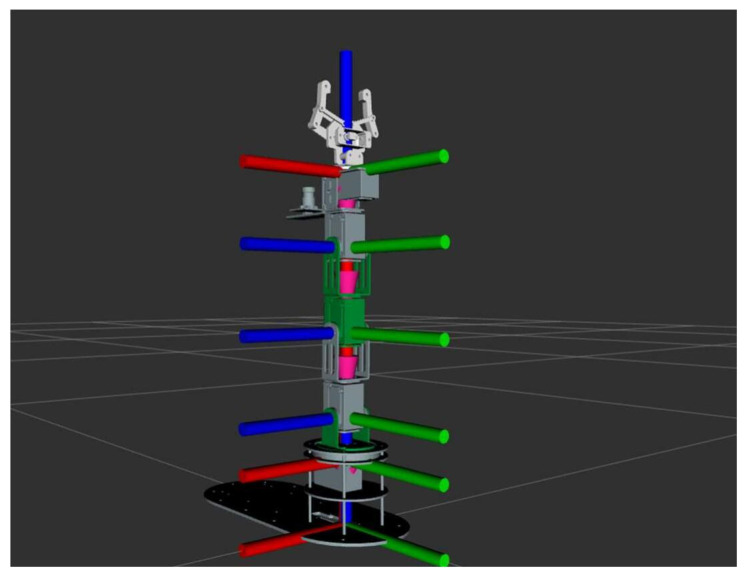
The picture of each coordinate system of Dofbot.

**Figure 16 sensors-24-00984-f016:**
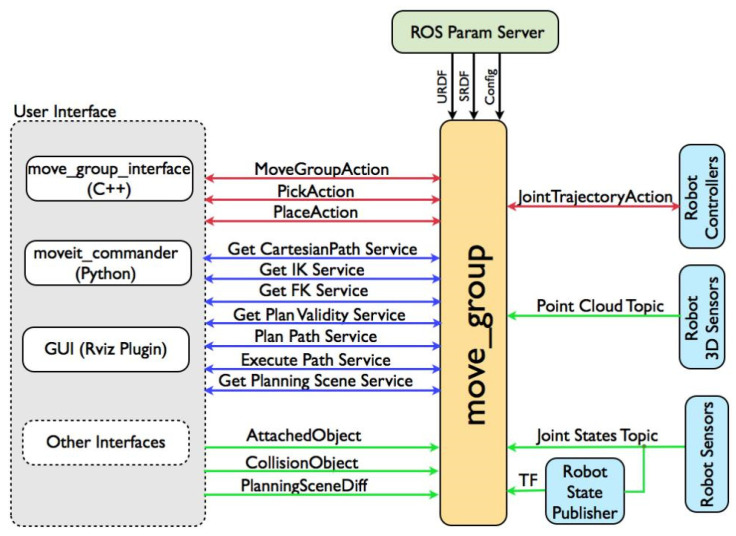
Move_group node system structure.

**Figure 17 sensors-24-00984-f017:**
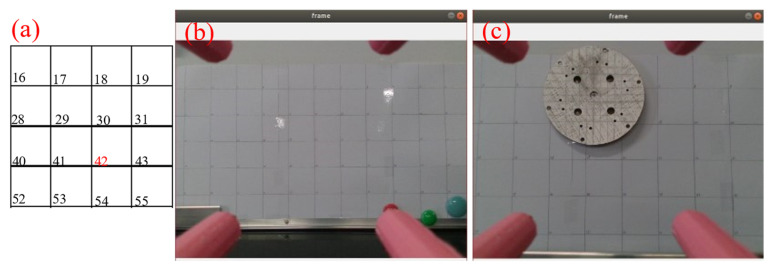
Disk tracking reality map: (**a**) grid lines in the neighborhood of point 42; (**b**) grid lines (each grid is 5 cm); (**c**) a porous disk with a diameter of 14 cm is moved on the magnetic plate with grid lines.

**Figure 18 sensors-24-00984-f018:**
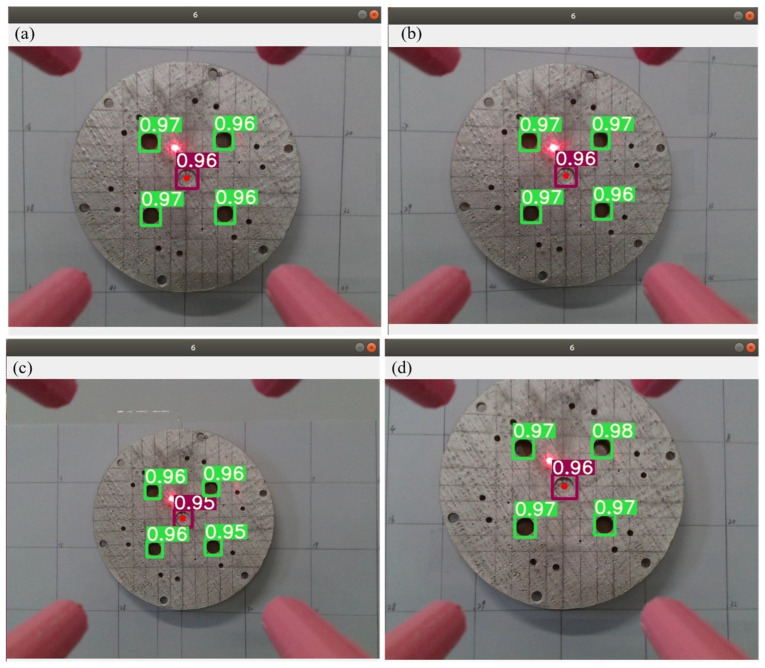
Dot 42 neighborhood trace: (**a**) at point 42, the laser landing point diagram; (**b**) move to 43; (**c**) move to 29; (**d**) move toward 30; (**e**) move toward 17; (**f**) move to 19; (g) move toward 31; (**h**) move to 41.

**Figure 19 sensors-24-00984-f019:**
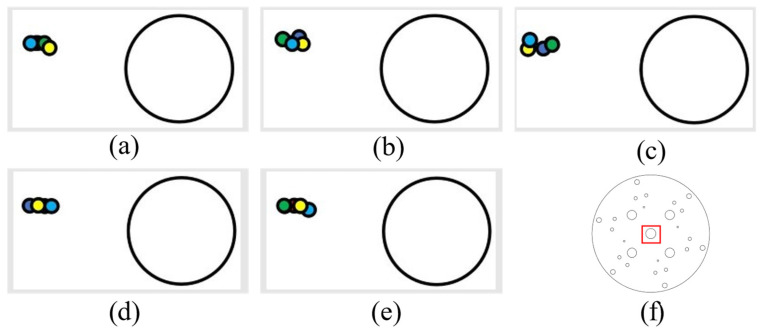
Left-moving laser drop-point maps: (**a**–**e**) are five sets of left-moving drop-point data maps, and (**f**) is the whole-disk map.

**Figure 20 sensors-24-00984-f020:**
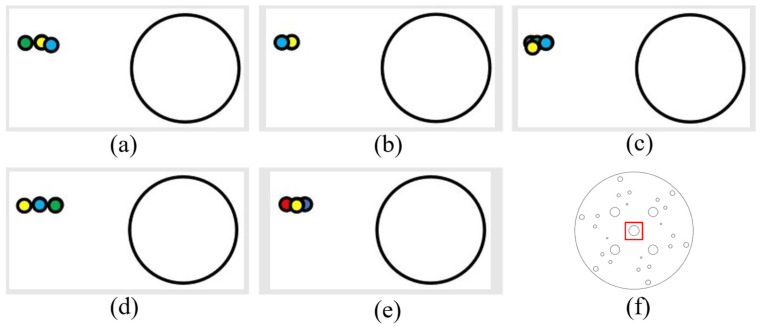
Right-moving laser drop-point maps: (**a**–**e**) are five sets of right-moving drop-point data maps, and (**f**) is the whole-disk map.

**Figure 21 sensors-24-00984-f021:**
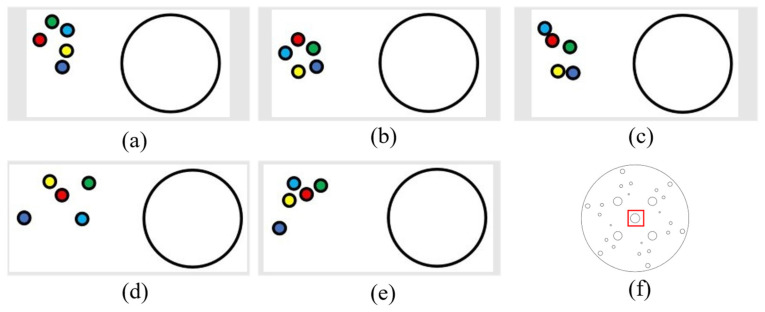
Upward-moving laser drop-point maps: (**a**–**e**) are five sets of upward-moving drop-point data maps, and (**f**) is the whole-disk map.

**Figure 22 sensors-24-00984-f022:**
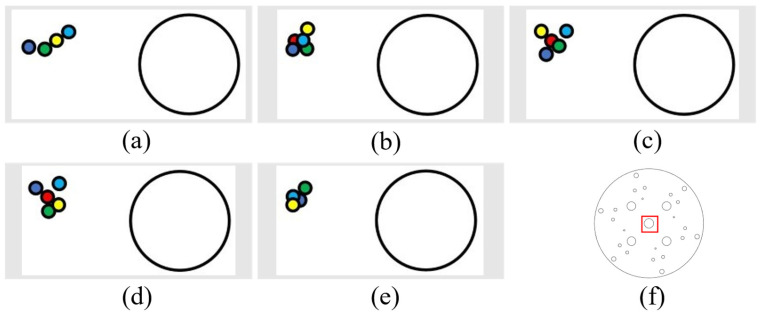
Downward-moving laser drop-point maps: (**a**–**e**) are five sets of downward-moving drop-point data maps, and (**f**) is the whole-disk map.

**Figure 23 sensors-24-00984-f023:**
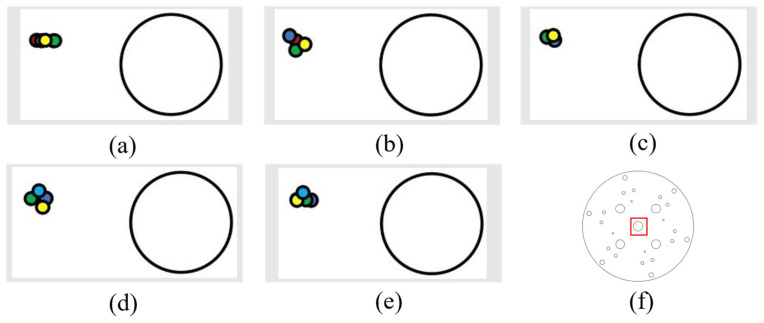
Left- and upward-moving laser drop-point maps: (**a**–**e**) are five sets of left- and upward-moving drop-point data maps, and (**f**) is the whole-disk map.

**Figure 24 sensors-24-00984-f024:**
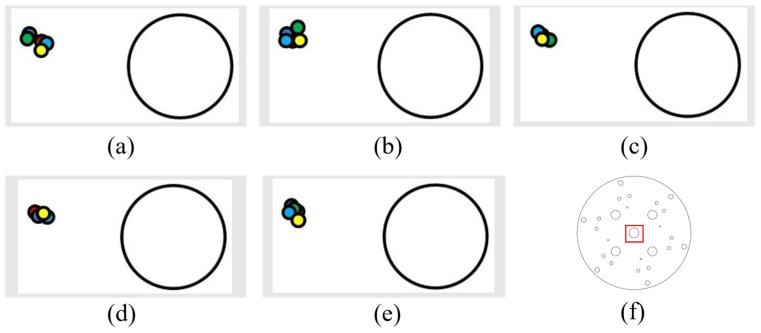
Right- and upward-moving laser drop-point maps: (**a**–**e**) are five sets of right- and upward-moving drop-point data maps, and (**f**) is the whole-disk map.

**Figure 25 sensors-24-00984-f025:**
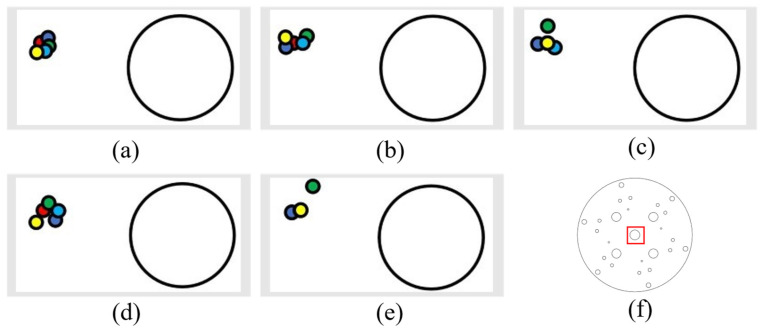
Left- and downward-moving laser drop-point maps: (**a**–**e**) are five sets of left- and downward-moving drop-point data maps, and (**f**) is the whole-disk map.

**Figure 26 sensors-24-00984-f026:**
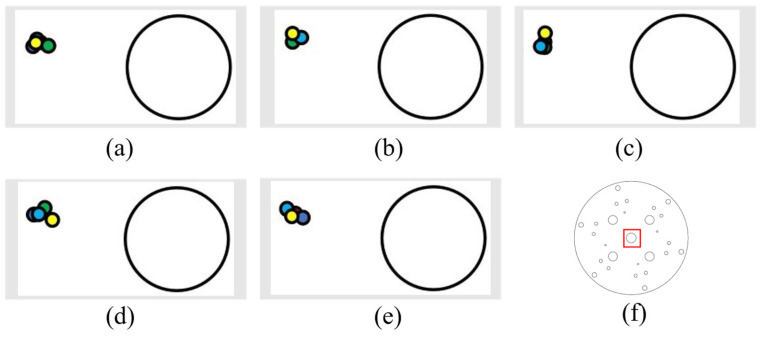
Right- and downward-moving laser drop-point maps: (**a**–**e**) are five sets of right- and downward-moving drop-point data maps, and (**f**) is the whole-disk map.

**Table 1 sensors-24-00984-t001:** Comparison of network structure for different models of YOLOv5.

	YOLOv5s	YOLOv5m	YOLOv5l	YOLOv5x
➀ Number of convolution kernel	32	48	64	80
➁ Number of convolution kernel	64	96	128	160
➂ Number of convolution kernel	128	192	256	320
➃ Number of convolution kernel	256	384	512	640
➄ Number of convolution kernel	512	768	1024	1280
Model depth multiple (depth_multiple)	0.33	0.67	1.0	1.33
Layer channel multiple (width_multiple)	0.5	0.75	1.0	1.25

**Table 2 sensors-24-00984-t002:** D-H parameters of Dofbot.

	$d_{i}$ (Along the z-axis)	$a_{i}$ (Along the x-axis)	$\beta_{i}$ (Around y-axis)	$\theta_{i}$ (Around z-axis)
1	0.066	0	0	$\theta_{1}$
2	0.04145	0	$\frac{\pi }{2}$	$\theta_{2}$
3	0	−0.08285	0	$\theta_{3}$
4	0	−0.08285	0	$\theta_{4}$
5	0	−0.07385	$-\frac{\pi }{2}$	$\theta_{5}$

## Data Availability

No new data were created or analyzed in this study. Data sharing is not applicable to this article.

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
