# Peer review of "Research on Multi-Hole Localization Tracking Based on a Combination of Machine Vision and Deep Learning"

_sensors, 2024, doi:10.3390/s24030984_

Round 1
Reviewer 1 Report
Comments and Suggestions for Authors
In this paper, the authors propose a novel method that combines machine vision and deep learning to control a robotic arm for the efficient and safe assembly of workpieces. The system described in the paper utilizes a camera to capture real-time images of targets in complex environments. These images are then processed and trained for recognition, enabling the detection and localization of porous disks. The paper is well written and contains quality research. The paper also acknowledges certain limitations. Firstly, the control accuracy of the servo was reported to be 1°, which was deemed inadequate as it led to the discarding of some values during positioning. Future research should focus on improving the control accuracy to enhance the precision of the assembly process. Secondly, the training performance of the model was relatively large, and the training time was relatively long. This suggests a need for further optimization of the machine learning algorithms and hardware conditions to reduce training time and improve overall system performance.
Some comments:
It is better to make the legend in Figures 4-5 a little larger.
Reviewer 2 Report
Comments and Suggestions for Authors
1. The contribution and novelty have to be clearly highlighted in the abstract.
2. The introduction or future work has to be enriched by related works:
https://doi.org/10.3390/pr11051507
3. The contributions have to be listed in points at the end of Introduction part.
4. The weaknesses of previous works have to be mentioned which have inspired this study.
5. The working principle (section 2.2) has to be supported by general block diagram.
6. The "Coordinate conversion" in Figure 2 has to be fused within the steam flow. It is isolated. Please, check this!.
7. The authors have excluded the dynamic of robot upon motion.
8. The characteristics of version for Yolov5 have to be explained within the same Table 1.
9. Figure 3 has been labeled, but it lacks from explanation.
10. Figure 4, 5, and 6 has low resolution and they have to be enriched by explanation.
11. Some figure like Figure 8 are common in robotic transformations. It has to be cited.
12. Please, Figure 8 and Figure 9 has to be fused in one Figure (with double subfigures).
13. The authors have neither cited equations, nor figures of section 4.1.
14. The "Calibration method" has to be supported by numeric example for more clarification. In addition, the "*" is not suitable for multiplication process.
15. Please, there is no need to use Figure for "LX-15D bus servos".
16. I have not seen the specification of Camera used in this study.
17. Figure 14 is badly labeled. In addition, the dimension of arms have to be given.
18. The authors have not mentioned the approach of control used in this study.
19. The subtitle "Robotic arm hardware configuration" has been repeated in both 5.2 and 5.3.
20. The pseudo code has to be presented. At least, part of the task used.
21. The results of real-time scenarios have not been shown!!!
22. Figure 16 does not make a sense.
23. The numerical results are very poor and the precisions of hole-identification has not be practically proved.
24. The conclusion is descriptive. It is void of quantitative and numerical improvement and comparison.
25. The future work has to be added.
Comments on the Quality of English Language
Minor editing of English language required
Reviewer 3 Report
Comments and Suggestions for Authors
This study has introduced an innovative approach that integrates machine vision and deep learning techniques for the precise localization of porous disks, subsequently facilitating the control of a robotic arm. The primary objective is to enhance operational efficiency and mitigate potential risks. The system outlined in this research leverages a camera to capture real-time images of targets within intricate environments. Subsequently, these images undergo training and processing using deep learning methodologies to achieve accurate recognition. The ultimate goal is to extract coordinate localization information, empowering the robotic arm to execute tasks with heightened precision and safety measures.
The manuscript is generally well-structured, providing a comprehensive set of experiments, including statistical and graphical simulations, offering a thorough evaluation of the proposed method.
While the writing is clear, refining the abstract for better clarity, providing more detailed explanations of the equations used, and expanding the conclusion section would enhance the overall coherence and completeness of the manuscript. Additionally, discussing limitations more thoroughly and providing information on code availability would contribute to the article's overall quality and impact. Minor grammatical and typographical errors are present throughout, suggesting the need for thorough proofreading.
Overall, the article presents a promising and potentially impactful contribution to the field, and addressing the suggested improvements would further strengthen its quality and significance.
Comments on the Quality of English LanguageMinor editing of English language required
Reviewer 4 Report
Comments and Suggestions for Authors
The addresses the manual assembly of parts in an industrial context, highlighting the limitations in terms of efficiency and worker risks associated with this process. The main goal is to improve assembly efficiency while reducing risks for human operators by introducing a system based on computer vision and deep learning. The system relies on the use of a camera to collect real-time images of parts to be assembled in complex environments. Deep learning techniques are applied to train algorithms capable of recognizing and precisely locating porous disks, a complex task due to the nature of the parts and possible variations. A camera captures images of parts in varied and complex industrial environments. The collected images are used to train deep learning models that can accurately recognize and locate porous disks. The obtained location information is converted into coordinates suitable for the robotic arm system through hand-eye calibration.
The robot receives these coordinates and is controlled to locate and track the holes on the parts to be assembled.
Results and Conclusions:
Although the method is promising, limitations may exist in terms of accuracy, reliability, and adaptability in real production environments. Improvements may be necessary to make the system more robust and better adapted to possible variations.
1. Implementation Complexity: Implementing a system using computer vision and deep learning requires advanced technical skills. The complexity of installation and configuration can pose challenges, especially for businesses or operators less familiar with these technologies.
2. High cost: The equipment required for this system, including high-resolution cameras, powerful computers and specialized software, represents a considerable financial investment. The initial cost may be prohibitive for some businesses.
3. Dependence on accuracy and data quality: The accuracy of recognition and localization of porous disks is highly dependent on the quality of the images collected and the training process of the algorithms. Changing environmental conditions or part defects can affect system accuracy.
4. Maintenance and reliability: Computer vision systems require regular maintenance to ensure their proper functioning. Additionally, their reliability can be compromised in the event of a breakdown or technical failure, leading to potential production interruptions.
5. Adaptability in real environments: Although the method has shown promising results under experimental conditions, its effectiveness in real production environments with unforeseen variations, obstacles or unanticipated defects on parts remains to be proven.
6. Technology Limitations: Systems based on computer vision and deep learning may have inherent limitations. For example, varying lighting conditions or atypically shaped rooms could affect system performance.
7. Operational Complexity: Integrating this system into an existing production process may be complex and require significant operational adjustments, which could result in temporary disruptions.
In conclusion, although the method offers potential benefits in terms of efficiency and risk reduction, it presents significant challenges in terms of implementation complexity, high costs, dependence on data accuracy, maintenance and reliability, as well as adaptability in real production environments. Significant improvements to overcome these challenges would be necessary for successful large-scale application
Comments on the Quality of English LanguageExtensive editing of English language required
Round 2
Reviewer 2 Report
Comments and Suggestions for Authors
The authors have addressed all my concerns. Thank you